# The Role of Liquid Biopsies in Detecting Molecular Tumor Biomarkers in Brain Cancer Patients

**DOI:** 10.3390/cancers12071831

**Published:** 2020-07-08

**Authors:** Heena Sareen, Celine Garrett, David Lynch, Branka Powter, Daniel Brungs, Adam Cooper, Joseph Po, Eng-Siew Koh, Joey Yusof Vessey, Simon McKechnie, Renata Bazina, Mark Sheridan, James van Gelder, Balsam Darwish, Mathias Jaeger, Tara L. Roberts, Paul De Souza, Therese M. Becker

**Affiliations:** 1Centre for Circulating Tumour Cell Diagnostics and Research, Ingham Institute for Applied Medical Research, 1 Campbell St, Liverpool, NSW 2170, Australia; h.sareen@student.unsw.edu.au (H.S.); celinegarrett@gmail.com (C.G.); 18292682@student.westernsydney.edu.au (D.L.); branka.powter@inghaminstitute.org.au (B.P.); adam.cooper@health.nsw.gov.au (A.C.); joseph.po@health.nsw.gov.au (J.P.); EngSiew.Koh@health.nsw.gov.au (E.-S.K.); jyus3681@uni.sydney.edu.au (J.Y.V.); tara.roberts@westernsydney.edu.au (T.L.R.); paulds@uow.edu.au (P.D.S.); 2South Western Sydney Clinical School, University of New South Wales South Western Clinical School, Goulburn St, Liverpool, NSW 2170, Australia; m.jaeger@unsw.edu.au; 3Western Sydney Clinical School, Western Sydney University, School of Medicine, Campbelltown 2560, NSW, Australia; 4School of Medicine, University of Wollongong, Wollongong, NSW 2522, Australia; daniel.brungs@health.nsw.gov.au; 5Liverpool Hospital, Elizabeth St & Goulburn St, Liverpool, NSW 2170, Australia; simonmckechnie@bigpond.com (S.M.); renata.bazina@health.nsw.gov.au (R.B.); marksheridan@bigpond.com (M.S.); james.vangelder@gmail.com (J.v.G.); balsam.darwish@brainsurgeon.net.au (B.D.); 6Department of Neurosurgery, Wollongong Hospital, Wollongong, NSW 2500, Australia

**Keywords:** glioma, biomarker, IDHI, MGMT, EGFR

## Abstract

Glioblastoma multiforme (GBM) is one of the most lethal primary central nervous system cancers with a median overall survival of only 12–15 months. The best documented treatment is surgical tumor debulking followed by chemoradiation and adjuvant chemotherapy with temozolomide, but treatment resistance and therefore tumor recurrence, is the usual outcome. Although advances in molecular subtyping suggests GBM can be classified into four subtypes, one concern about using the original histology for subsequent treatment decisions is that it only provides a static snapshot of heterogeneous tumors that may undergo longitudinal changes over time, especially under selective pressure of ongoing therapy. Liquid biopsies obtained from bodily fluids like blood and cerebro-spinal fluid (CSF) are less invasive, and more easily repeated than surgery. However, their deployment for patients with brain cancer is only emerging, and possibly suppressed clinically due to the ongoing belief that the blood brain barrier prevents the egress of circulating tumor cells, exosomes, and circulating tumor nucleic acids into the bloodstream. Although brain cancer liquid biopsy analyses appear indeed challenging, advances have been made and here we evaluate the current literature on the use of liquid biopsies for detection of clinically relevant biomarkers in GBM to aid diagnosis and prognostication.

## 1. Introduction

Glioblastoma multiforme (GBM) is the most common primary malignant brain tumor in adults, accounting for 62% of all brain tumors in Australia in 2013 and has a poor prognosis with a five-year survival of 4.6% and median overall survival (OS) estimates of 12–15 months [1]. GBM can be classified into two categories: primary (arises de-novo) or secondary (transforms from a previous lower grade tumor) and although they differ in terms of molecular characterization, treatment strategies and disease outcome overlap considerably [2]. The current standard of therapy for GBM comprises surgical resection, adjuvant radiotherapy, and concomitant chemotherapy with the alkylating agent, temozolomide, then adjuvant temozolomide where feasible [3]. The aggressiveness of GBM is thought to be due, in large part, to its treatment resistance caused by the presence of oncogenic mutations, ambiguous surgical margins and the blood-brain barrier which limits the uptake, and hence efficacy of systemic therapy [4]. In many cases, the efficacy of treatment is difficult to determine, particularly early during therapy, because therapy-associated tissue inflammation often resembles the effects of disease progression under magnetic resonance imaging (MRI); a phenomenon termed pseudoprogression [5]. As a result, determining true progression of disease is challenging, which in turn impacts on timely responses to treatment failure in patients. This issue has been a particular challenge in clinical trials, where no reliable surrogate marker is available for overall survival. It is clear that better biomarkers are needed to aid in the diagnosis and tracking of the clinical course of patients with GBM.

A non-invasive longitudinal approach for the diagnosis, prognostic assessment, molecular stratification, prediction of treatment response, and assessment of tumor progression is required for GBM. Circulating biomarkers are an appealing potential solution to this challenge. Circulating tumor cells (CTCs) and circulating tumor nucleic acids (ctNAs), and the molecular biomarkers that can be screened from these tumor entities, already served some of these roles in a number of other solid cancers [6]. In regards to brain cancer, liquid biopsies are challenging as the presence of the blood–brain barrier (BBB) impedes the release of tumor entities into the blood stream. The integrity of the BBB may however be compromised especially in advanced GBM. Although, for a long time, BBB was thought to prevent the release of CTCs and potentially ctNAs into the blood stream, recent studies showed that CTCs could be detected in patients with high-grade (WHO grade III and IV) brain cancers [7]. Other studies have also demonstrated that ctDNA and ctRNA can be detected in plasma from GBM patients as detailed below [8].

In patients with GBM, the lack of readily accessible tumor samples in some patients, and the lack of repeated debulking benefit in others, means simpler methods of assessing biomarkers from non-surgical samples warrant closer investigation. Here, we review potential biomarkers that may be relevant for GBM patient management, and whether they can be detected via liquid biopsies.

## 2. Molecular Biomarkers in Brain Cancer

A range of molecular biomarkers are associated with prognosis and may potentially stratify patients for different treatment strategies [9]. Key biomarkers for stratifying patients include IDH1 mutation, MGMT methylation, EGFRvIII mutation and/or EGFR amplification, GFAP mutation, hTERT promoter alterations, and loss of heterozygosity in chromosome 10 (LOH) (see Table 1). All of these biomarkers have been detected in CTC or ctNA assays.

### 2.1. Glial Fibrillary Acidic Protein (GFAP)

Glial fibrillary acidic protein (GFAP), an intermediate filament protein expressed by astrocytes and other central nervous cells is detected at significantly higher levels in GBM tissue compared to other intracranial lesions [10]. However, GFAP in serum cannot be used as a specific diagnostic measure for GBM due to the ‘sensitivity gap’ caused by heterogeneous/low expression of GFAP on some tumors that leads to the undetectable levels of GFAP released into the blood stream [28]. Higher serum GFAP levels correlate with tumor volumes, intra-tumoral GFAP expression, and extent of necrosis [28,29]. Serum GFAP levels associated with primary and recurrent high-grade glioma (HGG) tumor volumes and short PFS progression free survival (PFS). Higher preoperative GFAP serum levels also correlated with increased tumor volume and necrosis [30]. Furthermore, serum GFAP levels are associated with IDH1 mutational status (an established prognostic marker discussed below), with significantly lower serum GFAP found in IDH1 mutated HGGs [30]. GFAP is currently the most prevalent marker for the identification of GBM CTCs and expression is frequently maintained in GBM, despite its heterogeneity.

### 2.2. Methylguanine-DNA Methyltransferase Promoter Methylation (MGMT)

The O-6 methylguanine-DNA methyltransferase (MGMT) protein is involved in DNA repair, by reversing DNA alkylation [12]. Consequently, MGMT expression is linked to resistance to DNA alkylating agents, such as temozolamide, the main chemotherapeutic agent used for GBM [12]. Reduced MGMT expression, commonly caused by MGMT promoter methylation, renders cells more susceptible to temozolamide [31]. MGMT promoter methylation is more common in secondary GBM (75%) compared to primary GBM (26%) [32], and is associated with longer OS and PFS in patients treated with temozolamide and standard dose of radiotherapy (median OS of 22–26 months vs. 12–15 months in non-MGMT methylated tumors). Since temozolamide has toxicities, especially in patients with pre-existing comorbidities, detection of MGMT promoter methylation status may help tailor the best temozolamide dosage or schedule for the patient [31,33]. MGMT promoter methylation has been detected in plasma ctDNA of glioma patients, and methylation status correlated with GBM tissue, suggesting that liquid biopsy material could have potential in detecting MGMT promoter methylation status [13].

### 2.3. Isocitrate Dehydrogenase Mutations (IDH1/2)

Isocitrate dehydrogenase 1 and 2 (IDH1/2) enzymes catalyze the reversible oxidation of isocitrate to yield α-ketoglutarate with simultaneous reduction of NADP+ to NADPH. NADPH provides a cellular defense against intracellular oxidative damage [34,35]. The chemo-sensitivity of mutant IDH1 tumors is attributed to the impairment of DNA repair function resulting in increased DNA damage inducing apoptotic cell death [36]. About 12% of GBM patients carry mutations in IDH1 or IDH2; 90% of those carrying an IDH1 mutation have the specific R132H change, a missense mutation switching the amino acid arginine to histidine at position 132. The most common IDH2 mutation is IDH2-R172H [37].

IDH1 mutation rate reported in secondary GBM is 73–85%, whereas its rarely present in primary GBM [15,32,38]. IDH-mutant status is associated with longer OS in patients with WHO grade II-IV glioma. However, the association of higher rates of total surgical resections in IDH1 mutated malignant astrocytoma due to the clinical factors such as younger age, frontal location, and a non-enhancing disease component in the tumor mass may also contribute to better OS [39]. In vitro studies have shown that IDH1 mutant cells are more sensitive to radiation therapy as compared to wild type cells and low grade and secondary IDH1 mutant gliomas show increased chemosensitivity [40,41,42]. IDH mutations are therefore considered a positive prognostic marker for survival in grade II to IV gliomas [43].

IDH1 mutation detection was confirmed in plasma ctDNA of 80 glioma patients with 100% specificity and 60% sensitivity [16]. IDH1 mutations can be detected using various techniques such as pyrosequencing, immunohistochemistry and droplet digital polymerase chain reaction (ddPCR). ddPCR detection of the IDH1-R132H mutation is highly sensitive and suited for single cells and ctDNA analysis [44].

### 2.4. Epidermal Growth Factor Receptor (EGFR)

Epidermal growth factor receptor (EGFR) is a potential GBM biomarker. In normal cells, EGFR is involved in growth factor signaling, while cancer-associated oncogenic changes (mutations, overexpression, variant expression) often confer ligand independent oncogenic activity [45]. In brain cancer, one of the most widely investigated EGFR alterations is the EGFR transcript variant III (EGFRvIII), caused by varying DNA deletions in the gene that all affect mRNA splicing to exclude exons 2–7 [19]. Up to 33% of GBM express EGFRvIII, which has been associated with decreased survival, particularly in adolescents [19,46,47]. EGFRvIII is implicated in the process of gliomagenesis and in conferring resistance to chemotherapy [48]. Despite its proposed role in tumorigenesis, the prognostic significance of this mutant is still controversial. EGFRvIII overexpression in the presence of EGFR amplification was proposed as the strongest indicator of poor prognosis and survival [47]. In contrast, other studies suggest that EGFRvIII may be a positive prognostic marker and indicate prolonged survival of EGFRvIII patients treated with surgery and chemo/radiation therapy [49,50]. Regardless of this controversy, due to its prevalence, EGFRvIII detection in a pathological setting can help to clearly classify GBM. While it may also present a promising therapeutic target for GBM treatment, EGFR targeting tyrosine kinase inhibitors (TKIs, e.g., gefitinib and erlotinib) have not shown promise in clinical trials of GBM patients so far, possibly due to the poor penetration of these drugs through the BBB, thereby limiting the concentration of drug reaching the tumors [51]. Studies have shown that gliomas lack mutations in the EGFR exons 19–21 encoding the tyrosine kinase domain (common activating mutations in lung cancer that sensitize those cancers for gefitinib and erlotinib), and may be less dependent on EGFR kinase activity overall, also contributing to the failure of EGFR-TKIs [51]. Second generation EGFR-TKIs (afatinib and dacomitinib) have shown activity in GBM, but more studies are needed to confirm clinical utility. The BBB-permeable third generation inhibitor AZD9291 (osimertinib) may be an attractive candidate for EGFR inhibition therapy in GBM. Recent studies have shown that AZD9291 can significantly inhibit tumor growth and prolong animal survival in an orthotopic GBM model [52].

In lung cancer patients, EGFR mutations are readily detectable in liquid biopsies as an alternative to tissue biopsy [21]. While there are fewer such studies in brain cancer, DNA deletions causing EGFRvIII expression were detected in ctDNA of three of thirteen GBM patients in one study [19]. In this study, the exact deletions were first determined from matching tumor tissue genomic DNA by long range PCR, then primers adjacent to the deletions were generated to confirm presence in ctDNA. Thus, although, long range PCR assay is not useful for ctDNA, which has an average length of only 170 bases [19], the study confirmed detectability in ctDNA and confirmed the association of ctDNA amount with disease status (degree of total resection). As the exact deletions vary; detection assays from ctRNA would need to be further developed for routine EGFRvIII testing. Alternatively, an optimized GBM CTC enrichment protocol may allow to successfully detect EGFRvIII transcripts in CTC samples, comparable to androgen receptor variant 7 (AR-V7) analysis in prostate cancer [53].

### 2.5. Telomerase Promoter Mutations (TERT)

Telomerase reverse transcriptase (TERT) is a ribonucleoprotein enzyme essential for the replication of telomeres, the chromosome termini. Telomeres contain repetitive DNA sequences that become progressively shorter during successive cell divisions, ultimately leading to a permanent proliferative arrest. Telomerase is the main enzyme that counteracts telomere shortening through cell division and is normally only expressed in stem cells and gametes, but it is also central to cell transformation and immortalization during cancer development [54,55].

Two specific point mutations in the promoter of TERT (pTERT), C228T, and C250T (numeration relating to ATG start codon), have been identified in cancer cells and are proposed to activate telomerase [56]. These promoter mutations appear mutually exclusive [57]. pTERT mutations are common in many cancers and have been found in various cancers including GBM while not found in normal cells [54,56,58,59,60,61]. A high percentage of GBM samples (80–90%), have pTERT mutations correlated with increased TERT gene protein and accumulation. Patients with pTERT mutations had shorter OS than those without, 11 vs. 20 months, respectively [57,59,62,63].

pTERT mutations have previously been detected in liquid biopsy of patients with other cancers, and due its high prevalence, pTERT mutations detected from liquid biopsy would be predicted to be feasible in GBM patients and detection could contribute to future diagnostic assays [61,64].

### 2.6. Loss of Heterozygosity

Loss of heterozygosity (LOH) is the loss of genetic material from one of the two alleles of certain genes and is a frequently occurring genetic event in glioblastomas. LOH of 10q is found in both primary and secondary GBM occurring at the frequencies of 60–80% [2]. Complete loss of chromosome 10 has been exclusively associated with primary GBM [26]. The three commonly deleted loci on chromosome 10 are 10q14-p15, 10q23-24 (*PTEN*) and 10q25-pter [26]. The most important loss among these three is the loss of tumor suppressor gene *PTEN* along with other genes including *DMBT1, MXI1, LGI1, WDRI1,* and *FGFR2.* The PTEN protein is a phosphatase that plays an important role in inhibiting the PI3K/AKT/mTOR pathway. *PTEN* mutations or loss favor tumor development and loss of the *PTEN* loci 10q25-pter is associated with the progression of low-grade brain tumor and anaplastic astrocytoma to high grade glioblastomas.

LOH on chromosome 22 has also been found in GBM. The most frequent loss is that of chromosome 22q, which is found in 82% of secondary GBM and 41% of primary GBM. Deletion of the locus 22q12.3 leads to loss of the tumor suppressor gene *TIMP-3*, encoding the tissue inhibitor of metalloproteinases-3 (TIMP-3) [65]. TIMP-3 induces apoptosis and inhibits tumor cell growth and cancer progression. Chromosome 19q LOH is more frequently detected in secondary GBM (54%) than primary GBM (6%). LOH on chromosome 1 is often associated with longer survival but is a rare genetic event in both primary (12%) and secondary GBM (15%) [66]. Studies have also shown that LOH of 1p in combination with 19q is significantly associated with longer overall survival in glioblastoma patients [66,67].

LOH have been detected in the ctDNA of patients suffering from gliomas and other cancers [8,27]. Highly sensitive assays should be designed for the detection of LOH from liquid biopsies to facilitate the diagnosis and molecular subtyping of gliomas.

## 3. Liquid Biopsies

Liquid biopsies are known to carry a variety of entities, other than cells and DNA, shed by primary or metastatic lesions. In brain cancer, the main liquid biopsies that may be analyzed for tumor-specific biomarkers are blood and cerebro-spinal fluid (CSF) (Figure 1).

The schematic indicates benefits (green) and limitations (red) of (**a**) tissue biopsy vs. (**b**) liquid biopsies: cerebro-spinal fluid (CSF) and blood. A variety of entities including CTCs, ctNAs, exosomes, microRNA, proteins (proteomics), lipids (lipidomics), and metabolized products (metabolome) are available for analysis.

### 3.1. Circulating Tumor Cells (CTCs)

Circulating tumor cells (CTCs) are cells shed from primary or metastatic tumors into the blood stream [68]. CTCs are validated prognostic biomarkers for a variety of cancers such as lung, melanoma, osteosarcoma, pheochromocytoma, and parathyroid [69]. Importantly, CTCs can be a surrogate of tumor tissue and analyzed for the presence of molecular biomarkers [70]. The most commonly employed CTC detection methods rely on the presence of epithelial cell adhesion molecule (EpCAM), which is expressed on most carcinoma cell surfaces, but not on GBM cells [71]. Consequently, other strategies have been employed to detect GBM CTCs. Currently, a limited number of studies have detected CTCs in 21–82% of GBM patients by applying different methods for CTC enrichment and identification (see Table 2).

Studies demonstrate that GBM CTCs have a high mesenchymal and low pro-neural signature, with elevated EGFR copy number and chromosome alterations (gain of chromosomes 3, 7, and 12; and loss of 10, 13, and 22) comparable with their primary tumors. Mutational analysis of GBM CTCs using next generation sequencing (NGS) or more sensitive targeted approaches may help to stratify patient with high risk of relapse and direct treatment strategies [70,76]. Thus, CTC analysis may facilitate molecular subtyping of GBM, and as such may serve as prognostic and predictive biomarkers. Furthermore, the mesenchymal transcript expression in GBM CTCs suggests that a process similar to epithelial to mesenchymal transition (EMT) might modulate GBM homeostasis and dissemination. Therefore, CTC evaluation may provide unique biological insights into the relatively unknown pathogenesis and pathophysiology of GBM. Moreover, CTC monitoring may have the capacity to differentiate between tumor recurrence and pseudo-progression. There is some evidence that CTC counts reflect disease status, increasing with progression and falling in in response to chemo-radiotherapy [69]. Hence, CTC analysis could potentially complement conventional MRI to monitor GBM disease course.

Although CTCs have significant potential in their application to GBM, implementation into the clinical setting has a number of challenges. Firstly, CTC isolation from GBM patients appear to be limited by low CTC flux and method complexity [69]. Secondly, due to the heterogeneity of GBM, the information gained from the molecular subtyping and characterization of rare, individual CTCs may not be adequately representative of a patient’s entire GBM [71]. Thirdly, so far reports comprise small patient cohort sizes with no long-term data and challenging interpretations of any association with clinical outcomes to date [11]. Fourthly, it is worth considering that CTCs may have a limited role in monitoring the efficacy of the antiangiogenic agent bevacizumab, a standard systemic treatment used in GBM. As bevacizumab stabilizes the BBB, it may potentially mitigate the propagation of CTCs into systemic circulation and produce false negative results [71]; although better understanding of how CTCs traverse the BBB is needed. Finally, the question as to why GBM only rarely metastasizes, despite up to 82% patients carrying CTCs, remains to be answered.

### 3.2. Circulating Tumor Nucleic Acids (ctNAs)

Circulating tumor DNA and RNA (ctDNA and ctRNA, respectively, or ctNA in combination) are released into the bloodstream by breakdown of cancer tissue [77]. To analyze ctNA, the key challenge is to separate the signal from the noise by developing highly sensitive and specific methods to identify low concentrations of ctNA against the high background amounts of cell free DNA and RNA in the blood originating from normal cell homeostasis. Typically, this is achieved by detection of tumor-specific biomarkers, such as mutations like IDH1-R132H in GBM. In solid tumors, such as lung cancer, ctNA is more readily detected in patients with a higher tumor burden, and have been shown to be an early predictor of response to systemic treatment [21].

ctNA has been detected in patients with primary brain cancers including astrocytic (41) or oligodendroglial tumors (29). All oligodendroglial tumors and 80.5% of astrocytic tumors showed detectable biomarkers (MGMT promoter methylation and/or 10q LOH and/or 1p/19q LOH) in serum ctDNA in one study [8]. Serum ctDNA was used to detect methylation status of certain genes (MGMT, RASSF1A, CDKN2b, and CDKN2a) associated with the pathogenesis of CNS cancers in a cohort of 33 brain cancer patients (7 primary or recurrent GBM, 8 astrocytomas, 2 gliosarcoma, 6 meningiomas, and 10 other metastatic CNS cancer). Seventy percent of patients from glial tumor group had ctDNA-detectable promotor methylation of at least one gene. Similarly, at least one promotor was methylated in 7/10 patients of metastatic group and 3/6 patients who suffered from meningiomas [78]. As described above, various biomarkers relevant for GBM (IDH1, MGMT, GFAP, hTERT) are detectable in ctDNA from brain cancer patients, suggesting that the BBB does not effectively prevent its release into the blood. Nevertheless, there are challenges with the detection of ctDNA in GBM patient blood since concentrations of detected ctDNA is generally lower, and the proportion of ctDNA positive patients are less, compared to other cancers [79]. Highly sensitive techniques such as droplet digital PCR (ddPCR) assays or improved targeted next-generation sequencing (NGS) are emerging as successful approaches to identify and monitor ctDNA [80]. Since plasma is a readily accessible source of tumor tissue during a patient’s treatment journey, tracking ctNA changes may be a key tool to identify temporal molecular changes, improving the personalization of treatments for patients during different stages of their disease.

### 3.3. Cerebro-Spinal Fluid

CSF is a highly informative liquid biopsy source in GBM that can be analyzed for biomarker information. While clearly being more invasive than a peripheral blood draw, and carrying some risks not tolerated by all patients, it is also a less invasive procedure compared to surgical tissue biopsies. Longitudinal CSF evaluation during the disease course to monitor the disease progression may be possible, but patients would have to agree to repeated lumbar punctures. ctDNA can be isolated from CSF and current data suggest that it is comparably more abundant than plasma-derived ctDNA [81,82]. High-grade tumors (WHO grade III and IV) are more likely to have detectable ctDNA in CSF as compared to lower-grade tumors. For instance, 35 out of the 38 patients showed positive detection for pTERT mutation in their CSF sample with the specificity of 100% and sensitivity of 92.1% [23]. Not surprisingly, the sensitivity was much lower (7.9%) in matched plasma derived ctDNA samples, suggesting CSF is a superior liquid biopsy for detection of biomarkers. Critically, the study established significant correlation between the mutant allele frequency and disease burden and association with OS [23]; CSF ctDNA may thus be a prognostic marker. In another study, a custom FDA-authorized next generation sequencing assay was used to analyze the tumors of 85 GBM patients (WHO grade IV GBM: 46, grade III: 26, grade II: 13). For 49.4% (42/85) at least one tumor-derived genetic alteration was detected in CSF. For 20 patients, matched tissue biopsies and CSF was analyzed and all pairs shared mutations. CSF ctDNA also revealed copy number alterations (CNAs), promoter mutations, protein-coding mutations, and structural rearrangements correlating with poor survival rates, despite lack of association between ctDNA detection and tumor grade, disease duration, or prior therapy [82]. A study in primary lung cancer patients with secondary brain metastases supports the notion of CSF as a viable liquid biopsy source, as it compared NGS results from CSF ctDNA with plasma ctDNA and CTCs to find the mutation patterns of driver genes. The mutation detection rate was found to be 95.2% (20/21), 66.7% (14/21), and 39% (8/21) in CSF ctDNA, plasma ctDNA and CTCs, respectively. EGFR mutations were detected in 12 (57.1%) of patients via CSF as compared to only 5 patients in plasma ctDNA and blood-derived CTCs. EGFR mutation status of CSF-ctDNA was concordant with the EGFR mutation status of primary tumor in 88.9% (16/18) of patients [83].

Taken together, CSF may be a more reliable surrogate tumor source for biomarker testing than plasma, presumably due to its direct contact with the brain. However, while it has some potential to be used repeatedly and longitudinally to monitor disease progression, it is also more invasive. The trade-off between invasiveness and clinical acceptability on the one hand, with the accuracy of the method on the other therefore favors plasma-derived ctDNA as the first line approach. However, more data are needed to fully compare both sources, and it is possible that both can prove beneficial: initial mutation screening in CSF may reveal a mutation signature for each patient that later can be followed up in plasma ctDNA. Reports comparing plasma and CSF-derived ctDNA are summarized in Table 3.

### 3.4. Exosomes

Exosomes are membrane enclosed extra-cellular vesicles (EVs), generally 40–150 nm in diameter, that are actively released by both healthy cells and cancer cells. They carry various cell components such as proteins, nucleic acids (mRNA, DNA, non-coding RNA), and lipids. Docking onto other cells, they can exchange this cargo and thereby alter molecular activities in recipient cells. Exosomes released by cancer cells can be extracted as non-invasive, circulatory biomarkers that contain molecular characteristics of the original tumor and can be screened for the detection of these signatures. In one study, an orthotopic xenotransplant mouse model of human cancer stem cells showed that extracellular vesicles can cross the intact BBB and reach the bloodstream [85], suggesting that peripheral blood of both low- and high-grade glioma patients can be used to isolate EVs. These EVs were shown to be a source for detection of clinically relevant prognostic biomarkers, such as IDH1-R132H, and were successfully extracted from blood and CSF [85,86,87]. Higher exosome concentration in plasma of GBM patients compared to healthy individuals was demonstrated and linked to tumor recurrence in patients post-resection. Exosomes may be a potential biomarker to distinguish patients with GBM from not only healthy controls, but also from patients harboring other brain lesions [86,88], and may be helpful in the early diagnosis of disease [89]. Reports suggest that exosomal miRNA screening could be used as a predictive biomarker for GBM patients to monitor response to chemotherapy and drug resistance [90,91,92]. Further studies on larger cohorts are needed to validate exosome analysis as a diagnostic and therapeutic tool.

## 4. Conclusions

There is good emerging evidence that liquid biopsies, such as blood and CSF, can be used as potential surrogates for tissue biopsy for diagnostic and prognostic biomarker analysis in gliomas. However, in general, these studies are small, and do not provide sufficient statistical power for firm conclusions in regard to biomarker detection association with disease parameters. Given the relative difficulty of obtaining brain tissue, and the challenges associated with monitoring brain cancer and determining treatment response, improved strategies to develop superior biomarkers are essential. Liquid biopsies offer a more accessible source of molecular information, which may allow diagnosis and characterization without invasive surgery. The disrupted BBB, which is a hallmark of GBM, may offer a window into the biological behavior through the study of liquid biopsies. Despite generally reduced detection of plasma ctNA in brain cancer, monitoring of disease progression may still be useful with plasma ctDNA for individual patients, as long as specific tumor-associated mutations are known for the patient, potentially via initial CSF screening. CTCs where detectable, may give information regarding novel proteins expressed on cancer cells, such as PD-L1, that may be a prognostic predictor of OS and may possibly suggest alternative management strategies.

## Figures and Tables

**Figure 1 cancers-12-01831-f001:**
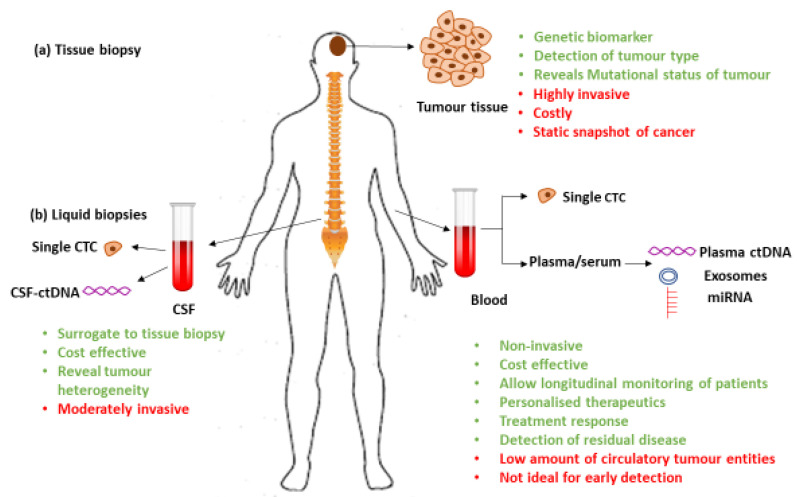
Comparison of tissue and liquid biopsies.

**Table 1 cancers-12-01831-t001:** Brain cancer biomarkers and detection in liquid biopsies.

Marker	Clinical Utility in Brain Cancer	Detected in Brain Cancer Circulating Tumor Cells (CTCs)	Detected in Brain Cancer ctNAs	Detected in CTCs or ctNAs #
**GFAP**	Yes [10]	Yes [11]	No	No
**MGMT ***	Yes [12]	No	Yes ^+^ [13]	Yes, colorectal cancer [14]
**IDH1**	Yes [15]	No	Yes ^+^ [16]	Yes, leukaemia [17]
**EGFR ****	Emerging [18]	No	Yes ^+^ [19]	Yes, lung cancer [20,21]
**hTERT**	Emerging [22]	Yes [7]	Yes ^+^ [23]	Yes, Urothelial cancer [24], Metastatic breast cancer [25]
**LOH chr10**	Yes [26]	No	Yes ^+^ [8]	Yes, Ovarian cancer [27]

LOH chr10 loss of heterozygocity chromosome 10; # in other cancers; * MGMT promoter methylation; ** EGFR mutations including variant III; ^+^ plasma derived circulatory tumour nucleic acids (ctNA).

**Table 2 cancers-12-01831-t002:** CTC detection in glioblastoma multiforme (GBM) patient blood.

CTC isolation/ID Method (Reference)	Patient No.	CTC Counts #	Efficiency *	Biomarker Tested	Clinical Utility	Limitations
Lentiviral telomerase reverse transcriptase (TERT)-promoter based detection [7]	11	8.8 (pre-RT)	pre-RT: 72% (8/11) post RT: 8% (1/11)	Epidermal growth factor receptor (EGFR) amplification	Prognostic marker (increased CTC count with recurrence)	Small cohort size, requires viral transduction limited to viable cells, may affect biomarker detection
Gradient PBMCs/CTC enrichment, immunocytostainig for glial fibrillary acidic protein (GFAP) [11]	141	0.1–2.2	20.6% (29/141)	EGFR amplification	No correlation with OS	GFAP heterogeneous in GBM, use as sole ID marker may under-estimate CTC counts
CTC-ichip leucocytes depletion, CTC ID with probing for SOX2, Tubulin b-3, EGFR, A2B5 and c-MET [72]	33	11.8 (progressive disease) 2.1 (stable disease)	39% (13/33)	N/A	Prognostic marker (progressive disease with greater frequency of CTCs)	Small cohort size
FISH CTC detection (chromosome 8 polyploidy) and exclusion of CD45^+^ cells [69]	31	0.13–1.33	71% (24/31)	N/A	CTC count decreases post adjuvant therapy, CTCs may help distinguish radio-necrosis from true tumor progression	Small cohort size
Parsortix platform for size based CTC capture, CTC ID testing by EGFR, Ki67, EB1 probing [73]	13	0.3, 1 patient: CTC clusters	53.8% (7/13)	NGS: *APC*, *XPO1*, *TFRC*, *JAK2*, *BRCA2*, *ERBB4* and *ALK*	N/A	Small cohort size
VAR2CSA malaria protein based targeted immunomagnetic isolation [74]	5 (GBM)	3.5~	80% (4/5)	NGS: IDHI, RB1, ALK, LOH 1p/19q, MGMT	N/A	Small cohort size, use of VAR2CSA for both isolation and ID may reduce specificity of CTC detection
MCAM, MCSP targeted immunomagnetic isolation, GFAP and GLAST probing [75]	13 (15 samples)	1.5	60% (9/15)	N/A	No correlation of CTC counts with PFS/OS	Small cohort size

ID CTC identification, # Average CTC counts detected normalised per 1 mL blood; * proportion of CTC positive patients; RT radiotherapy; N/A not applicable; ~estimated from graph in [74] for 5 GBM patients.

**Table 3 cancers-12-01831-t003:** Comparison of CSF and blood derived ctDNA.

ctDNA Source (Reference)	Patients No	Biomarker Tested	Percentage Detection #	Relevance to Disease
Plasma ctDNA, CSF ctDNA [81]	12	NGS: EGFR, PTEN, ESR1, IDH1, ERBB2, FGFR2	“higher” sensitivity of mutation detection from CSF ctDNA	Nd
Plasma ctDNA, CSF ctDNA [82]	85	NGS: TERT, TP53, IDH1, EGFR, and EGFRvIII	49.4% vs. 15.7% CSF vs. plasma ctDNA	Prognostic: shorter survival of CSF ctDNA positive patients
Plasma ctDNA, CSF ctDNA [23]	38	pTERT mutation	92.1% vs. 7.9% CSF vs. plasma ctDNA	pTERT mutation potential poor survival predictor
Plasma ctDNA, CSF ctDNA * [83]	21	NGS: EGFR, KIT, PIK3CA, TP53, SMAD4, ATM, SMARCB1, PTEN	95.2% vs. 66.7% CSF vs. plasma ctDNA	Nd
Plasma ctDNA, CSF ctDNA [84]	7	NGS: NF2, AKT, BRAF, NRAS, EGFR	CSF ctDNA detection “significantly higher” with low systemic disease burden	Nd

* Lung cancer metastasised to brain; # proportion of patients with detected ctDNA; Nd not determined.

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
