# Peer review of "The Role of Liquid Biopsies in Detecting Molecular Tumor Biomarkers in Brain Cancer Patients"

_cancers, 2020, doi:10.3390/cancers12071831_

Round 1

Reviewer 1 Report

Authors aimed to review the role of liquid biopsies in detecting molecular tumour biomarkers in brain cancer patients. Article is concisely written; conclusions are supported by the data. 

Major points 

-        none  

Minor essential revisions:

-        none

Author Response

Reviewer did not raise any issues.

Reviewer 2 Report

In the present manuscript the authors provide a comprehensive overview on liquid biopsies in glioblastoma multiforme (GBM). This fatal disease is characterized by a very poor outcome, which in part is due to the difficulty of monitoring treatment success and relapse. In this regard, liquid biopsies could have clinical relevance, as ctDNA, CTCs, and exosomes have been proven as useful in other cancer entities.

The authors give an overview on the most frequently used molecular markers in brain cancer, and secondly list the most relevant biomaterials to assess these marker which had been reported so far. The overview is concise but informative and addresses the potential pros and cons of each approach.

Author Response

Reviewer did not raise any issues.

Reviewer 3 Report

In this manuscript, Sareen et al have reviewed detection of molecular tumor biomarkers in brain cancer patients by liquid biopsies. They have summarized recent studies on the application of liquid biopsies to diagnose brain tumor (GBM) and the manuscript fits the scope of Cancers, but there are several issues to address for publication. Therefore, this manuscript has to be addressed the following issues.

Minor issues

  1. Please correct references (for example, [8]).

  1. As authors have mentioned, blood-brain barrier (BBB) is one of huddles to improve collection yield of ctNAs in blood or cerebro-spinal fluid (CSF). It will be worthy of describing BBB roles in brain tumorigenesis in this manuscript.

  1. It would be better to add recently published literatures and update references in the manuscript such as,

Cell Rep. 2020 Feb 18;30(7):2065-2074.e4. doi: 10.1016/j.celrep.2020.01.073.

Clin Cancer Res. 2019 May 15;25(10):3115-3127. doi: 10.1158/1078-0432.CCR-18-2946. Epub 2019 Jan 24.

Cancer Lett. 2018 Nov 1;436:10-21. doi: 10.1016/j.canlet.2018.08.004. Epub 2018 Aug 10.

 ..so on..

Author Response

We would like to thank you for your valuable comments/suggestions.

We amended the manuscript according to the suggestions. All changes in the manuscript are highlighted in yellow:

  1. Please correct references (for example, [8]).

In the Reference section reference 8 (page 11, lines 382-383) and 30 (page 12, lines 436-438) have been amended.

   2. As authors have mentioned, blood-brain barrier (BBB) is one of the hurdles to improve collection yield of ctNAs in blood or cerebro-spinal fluid (CSF). It will be worthy of describing BBB roles in brain tumorigenesis in this manuscript.

We agree in principal with the reviewer. The role of the BBB in tumorigenesis, metastasis and therapy resistance is complex and interesting. However, in depth reviewing of the BBB is beyond the scope of this biomarker/liquid biopsy focused review. We did add a small section to address the BBB role in regards of liquid biopsies.

lines 58-60 “In regards to brain cancer, liquid biopsies are challenging as the presence of the blood- brain barrier (BBB) impedes the release of tumour entities into the blood stream. The integrity of the BBB may however be compromised especially in advanced GBM.”

3. It would be better to add recently published literatures and update references in the manuscript such as,

Cell Rep. 2020 Feb 18;30(7):2065-2074.e4. doi: 10.1016/j.celrep.2020.01.073.

Clin Cancer Res. 2019 May 15;25(10):3115-3127. doi: 10.1158/1078-0432.CCR-18-2946. Epub 2019 Jan 24.

References 91 (page no-16 lines 580-582), 94-95 (page no-16 lines 587-591) have been introduced in the review and added to the reference section.

Reviewer 4 Report

The authors summarized the current literature of molecular biomarkers in the brain (GFAP, MGMT, IDH1/2, EGFR, TERT, LOH) and liquid biopsies(CTCs, ctNAs, Cerebro-spinal fluid, Exosomes), to evaluate the relevant biomarkers in GBM by using liquid biopsies.

This review provides a sound and meaningful reference for clinical diagnosis and prognostication.

Author Response

Reviewer did not raise any issues.